# Clinical gait analysis using video-based pose estimation: Multiple perspectives, clinical populations, and measuring change

**Jan Stenum**[1,2], **Melody M. Hsu**[1,3], **Alexander Y. Pantelyat**[4], **Ryan T. Roemmich**[1,2]*

**1** Center for Movement Studies, Kennedy Krieger Institute, Baltimore, Maryland, United States of America, **2** Department of Physical Medicine and Rehabilitation, The Johns Hopkins University School of Medicine, Baltimore, Maryland, United States of America, **3** Department of Neuroscience, The Johns Hopkins University School of Medicine, Baltimore, Maryland, United States of America, **4** Department of Neurology, The Johns Hopkins University School of Medicine, Baltimore, Maryland, United States of America

* rroemmi1@jhmi.edu

## Abstract

Gait dysfunction is common in many clinical populations and often has a profound and deleterious impact on independence and quality of life. Gait analysis is a foundational component of rehabilitation because it is critical to identify and understand the specific deficits that should be targeted prior to the initiation of treatment. Unfortunately, current state-of-the-art approaches to gait analysis (e.g., marker-based motion capture systems, instrumented gait mats) are largely inaccessible due to prohibitive costs of time, money, and effort required to perform the assessments. Here, we demonstrate the ability to perform quantitative gait analyses in multiple clinical populations using only simple videos recorded using low-cost devices (tablets). We report four primary advances: 1) a novel, versatile workflow that leverages an open-source human pose estimation algorithm (OpenPose) to perform gait analyses using videos recorded from multiple different perspectives (e.g., frontal, sagittal), 2) validation of this workflow in three different populations of participants (adults without gait impairment, persons post-stroke, and persons with Parkinson's disease) via comparison to ground-truth three-dimensional motion capture, 3) demonstration of the ability to capture clinically relevant, condition-specific gait parameters, and 4) tracking of within-participant changes in gait, as is required to measure progress in rehabilitation and recovery. Importantly, our workflow has been made freely available and does not require prior gait analysis expertise. The ability to perform quantitative gait analyses in nearly any setting using only low-cost devices and computer vision offers significant potential for dramatic improvement in the accessibility of clinical gait analysis across different patient populations.

## Author summary

People that experience a stroke or are diagnosed with Parkinson's disease often have mobility impairments such as slow walking speeds, shortened steps and abnormal movement of the legs during walking. It is a challenge for clinicians to measure and track the

**Data Availability Statement:** The dataset of unimpaired gait is available from http://bytom.pja.edu.pl/projekty/hm-gpjatk. The stroke and PD

datasets contain videos with identifiable information and are therefore not available. Code for our workflow is available at https://github.com/janstenum/GaitAnalysis-PoseEstimation/tree/Multiple-Perspectives.

**Funding:** We acknowledge funding from the NIH (grant R21 HD110686 to RTR), RESTORE Center Pilot Project Award (to RTR via NIH grant P2CHD101913), the American Parkinson Disease Association (grant 964604 to RTR), and the Sheikh Khalifa Stroke Institute at Johns Hopkins Medicine to RTR. The funders had no role in study design, data collection and analysis, decision to publish, or preparation of the manuscript.

**Competing interests:** The authors have declared that no competing interests exist.

multitude of walking parameters that can indicate recovery or progression of disease in an objective and quantitative manner. We present a new workflow that allows a user to analyze the gait pattern of a person walking recorded with only a single video obtained with a smartphone or other digital recording device. We test our workflow is 3 groups of participants: persons with no gait impairment, persons post-stroke, and persons with Parkinson's disease. We show that a user can perform these video-based gait analyses by recording videos with views from either the side or the front, which is important given the space restrictions in most clinical areas. Our workflow can produce accurate results as compared with a gold standard three-dimensional motion capture system. Furthermore, the workflow can track changes in gait, which is needed to measure changes in mobility over time that may occur because of recovery or progression of disease. This work offers potential for dramatic improvement in the accessibility of clinical gait analysis across different patient populations.

## Introduction

Walking is the primary means of human locomotion. Many clinical conditions–including neurologic damage or disease (e.g., stroke, Parkinson's disease (PD), cerebral palsy), orthopedic injury, and lower extremity amputation–have a debilitating effect on the ability to walk [1–3]. Quantitative gait analysis is the foundation for effective gait rehabilitation [4]: it is critical that we objectively measure and identify specific deficits in a patient's gait and track changes. Unfortunately, there are significant limitations with the current state-of-the-art. Marker-based motion capture laboratories are considered the gold standard measurement technique, but they are prohibitively costly and available largely to select hospitals and research institutions. Other commercially available technologies (e.g., gait mats, wearable systems) only provide predefined parameters (e.g., spatiotemporal data or step counts), are relatively costly, and require specific hardware. There is a clear need for new technologies that can lessen these barriers and provide accessible and clinically useful gait analysis with minimal costs of time, money, and effort.

Recent developments in computer vision have enabled the exciting prospect of quantitative movement analysis using only digital videos recorded with low-cost devices such as smartphones or tablets [5–7]. These pose estimation technologies leverage computer vision to identify specific "keypoints" on the human body (e.g., knees, ankles) automatically from simple digital videos [8,9]. The number of applications of pose estimation for human health and performance has increased exponentially in recent years due to the potential for dramatic improvement in the accessibility of quantitative movement assessment [6,7,10]. We have previously used OpenPose [8]–a freely available pose estimation algorithm–to develop and test a comprehensive video-based gait analysis workflow, demonstrating the ability to measure a variety of spatiotemporal gait parameters and lower-limb joint kinematics from only short (<10 seconds) sagittal (side view) videos of individuals without gait impairment [11]. Others have also used a variety of approaches to combine pose estimation outputs and neural networks to estimate different aspects of mobility [5,12–16].

This foundational work in using pose estimation for video-based gait analysis has demonstrated significant potential of this emerging technology. There are now prime opportunities to build upon what has already been developed and progress toward direct clinical applications. In moving toward clinical application, we considered the needs for: 1) flexible approaches that can accommodate different perspectives based on the space constraints of the

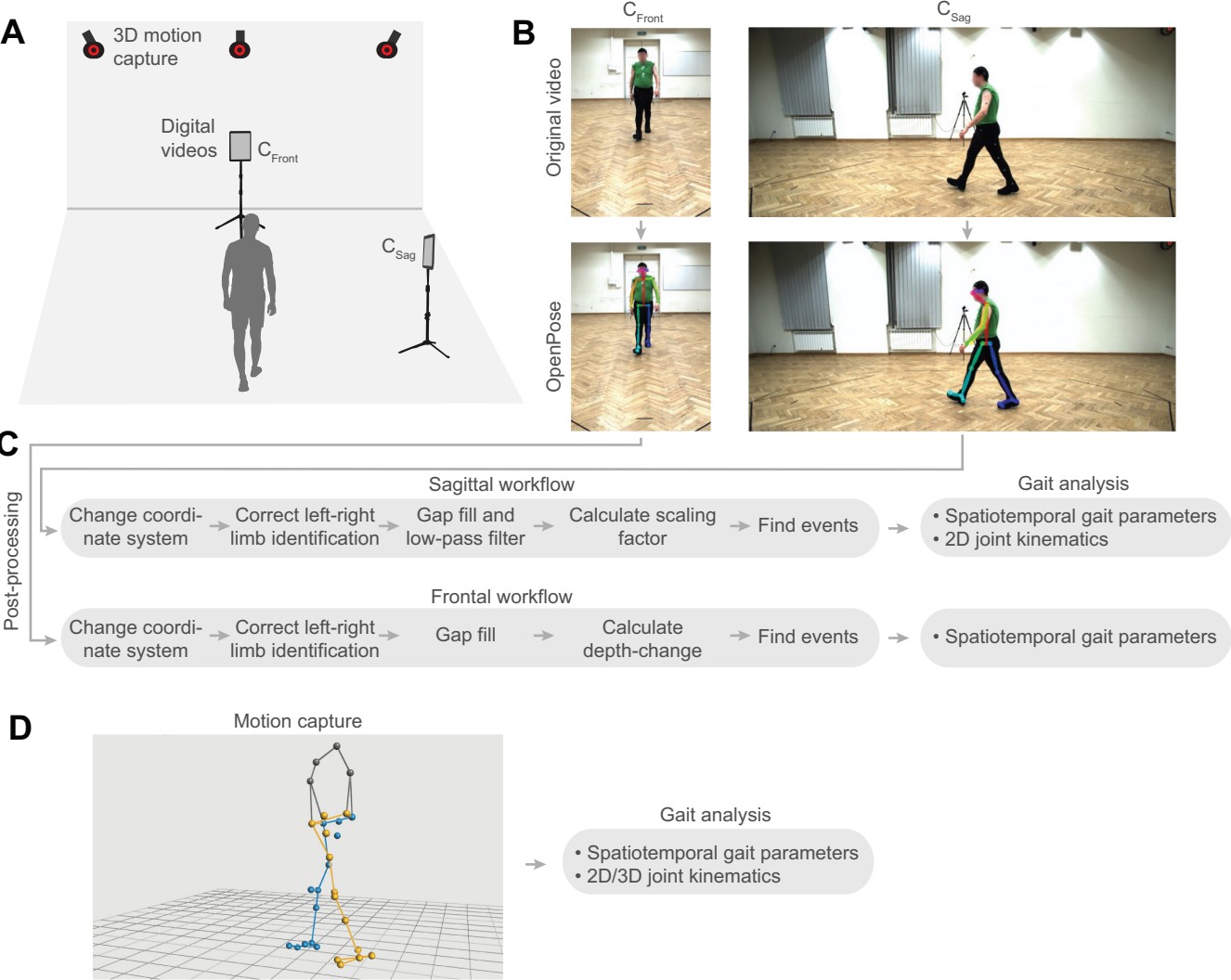

**Fig 1. Conceptual overview.** We recorded three-dimensional (3D) motion capture and digital videos of gait trials performed by persons post-stroke and persons with Parkinson's disease (A). We analyzed digital videos of the frontal (C_Front) and sagittal plane (C_Sag) with OpenPose to track anatomical keypoints (B). We developed workflows to perform a gait analysis, independently, for videos of the frontal and sagittal plane (C). See Methods section for detailed information about the frontal and sagittal plane post-processing workflows. Note that the 'Calculate depth-change time-series' step in the frontal workflow contains multiple sub-steps including tracking the pixel size of the torso and low-pass filtering (see S4 Fig for justification of tracking method and smoothing). We compared spatiotemporal gait parameters and joint kinematics from our workflows to parameters obtained with 3D motion capture (D).

end user (e.g., a clinician may only have access to a long, narrow hallway or hospital corridor where a sagittal recording of the patient is not possible), 2) testing and validation directly in clinical populations with gait dysfunction, 3) measurement of clinically relevant gait parameters that are of particular relevance to specific populations, and 4) the ability to measure changes in gait that occur in response to a change in speed.

Here, we present a novel, versatile approach for performing clinical gait analysis using only simple digital videos. First, we developed and tested a novel workflow that performs a gait analysis using frontal plane recordings of a person walking either away from or toward the camera (Fig 1). Our approach is based on tracking the size of the person as they appear in the video image (measured with keypoints from OpenPose) and using trigonometric relationships to estimate depth and, ultimately, spatial parameters such as step length and

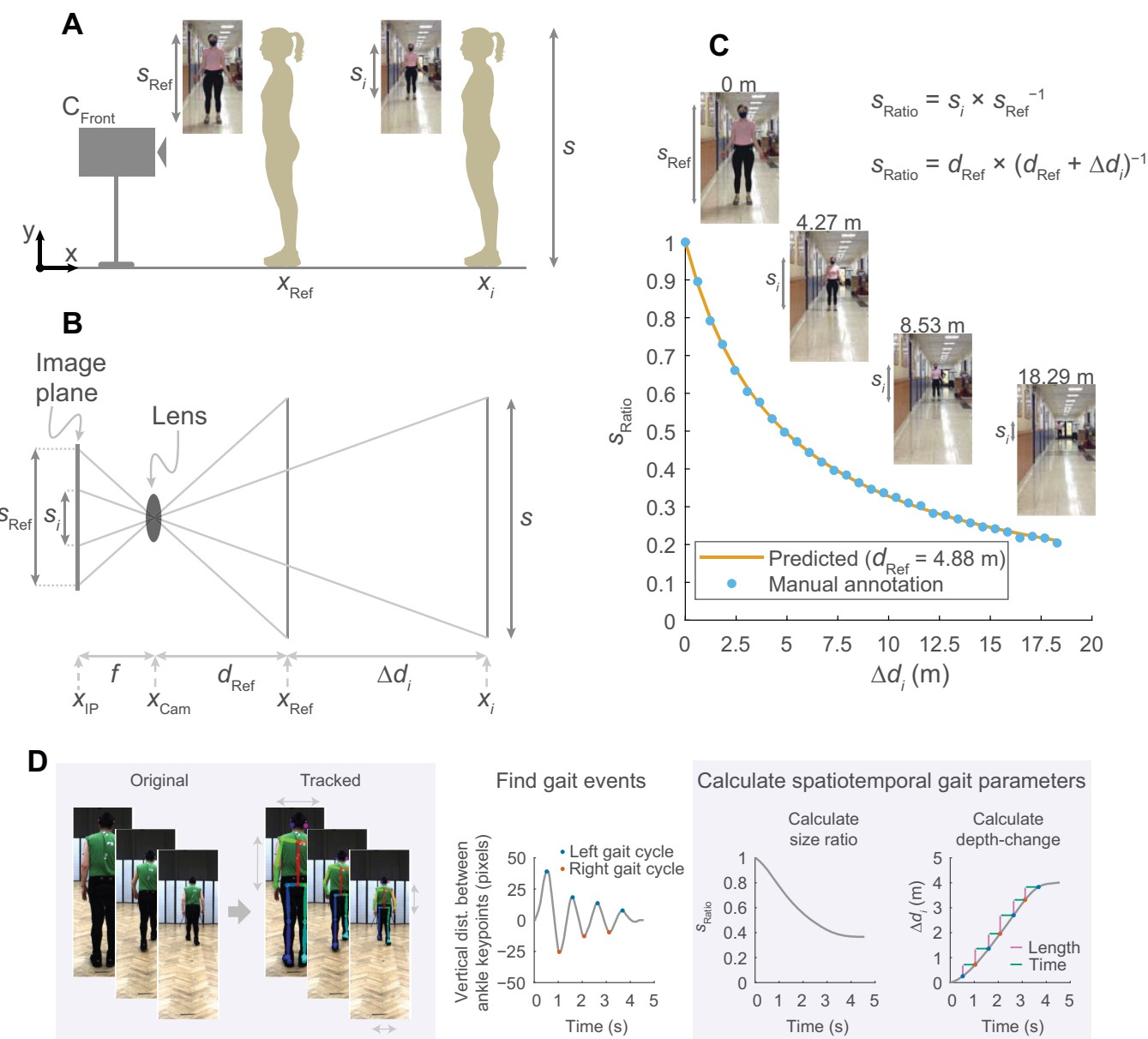

**Fig 2. Diagram of frontal plane analysis to obtain spatiotemporal gait parameters.** A person of size (height) $s$ stands at two distances from a frontal plane camera ($C_{Front}$; panel A): an initial reference depth ($d_{Ref}$) and at a depth-change ($\Delta d_i$). The size in pixels of the person at each depth are denoted by $s_{Ref}$ and $s_i$. From trigonometric relationships we derive a relationship between pixel size and depth-change (B, see Methods for detailed explanation; $f$, focal length of camera; $x_{IP}$, position of image plane of camera; $x_{Cam}$, position of camera lens; $x_{Ref}$, initial position of person; $x_i$, position of person following depth-change). The predicted pixel sizes of a person standing at increasing depths closely tracks manually annotated pixel sizes, which shows that we can use pixel size to estimate depth-changes (C). Summary of our frontal plane workflow (D): OpenPose tracks anatomical keypoints, we find gait cycle events, calculate a time-series of pixel size, and calculate depth-change at which point step lengths and step times can be derived.

gait speed (Fig 2; see expanded description in Methods). Second, we test both our frontal and sagittal workflows directly in two clinical populations with gait impairments that result from neurologic damage or disease (persons post-stroke or with Parkinson's disease).

## Results

### Development and testing of a novel approach for gait videos recorded in the frontal plane

We first validated our frontal plane approach during overground walking in a group of young participants without gait impairment (we have previously demonstrated the accuracy of obtaining gait parameters using sagittal plane videos in the same dataset of unimpaired participants [11]). We then compared spatiotemporal gait parameters (step time, step length and gait speed; averaged values for a single walking bout) simultaneously obtained with 3D motion capture and with frontal plane videos positioned to capture the person walking away from one camera and toward the other camera (data collection setup shown in Fig 3A).

Step time showed average differences (negative values denote greater values in video data; positive values denote greater values in motion capture data) and errors (absolute difference) up to one and two motion capture frames (motion capture recorded at 100 Hz; 0.01 and 0.02 s), respectively, between motion capture and frontal plane video (Fig 3B and S1 Table). The 95% limits of agreement between motion capture and frontal plane videos ranged from

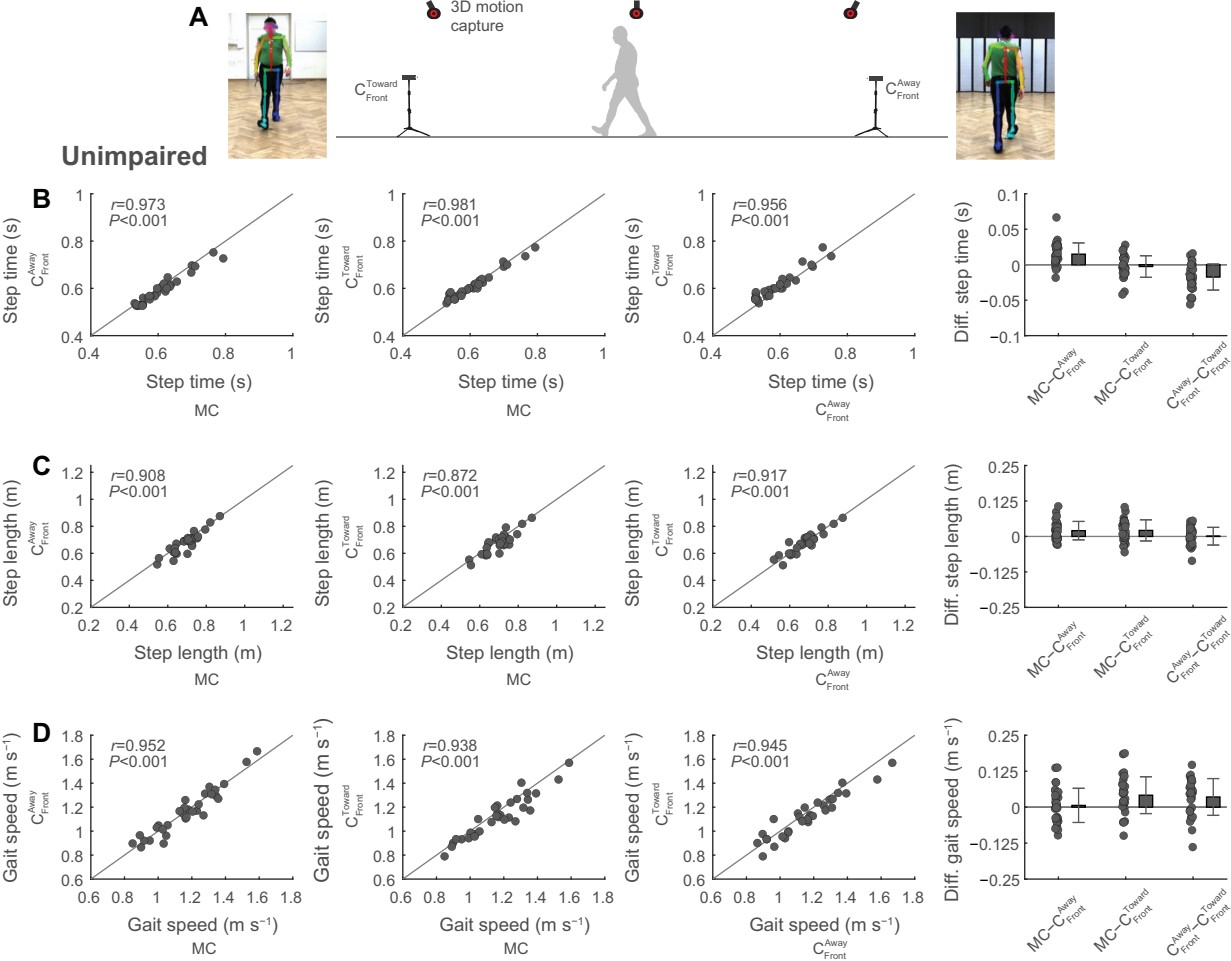

**Fig 3. Testing of a novel approach for spatiotemporal gait analysis from videos of unimpaired adults recorded in the frontal plane.** We recorded digital videos of the frontal plane where the person walked toward one camera and away from the other camera (A). We compared spatiotemporal gait parameters (B, step time; C, step length; D, gait speed) between the two digital videos and 3D motion capture (see S1 Table).

−0.03 to 0.05 s, suggesting that 95% of differences with motion capture fell within this interval. Step length showed average differences and errors up to about 0.02 and 0.03 m, respectively, between motion capture and frontal plane videos (Fig 3C). The 95% limits of agreement between motion capture and frontal plane videos ranged from −0.052 to 0.094 m. Gait speed showed average differences and error up to 0.04 and 0.06 m s$^{-1}$, respectively, with 95% limits of agreement ranging between −0.11 and 0.17 m s$^{-1}$ (Fig 3D). Correlations for all spatiotemporal gait parameters between motion capture and frontal plane videos were strong (all $r$ values between 0.872 and 0.981, all $P<0.001$; Fig 3B–3D).

## Testing of video-based gait analysis in persons with neurologic damage or disease

Next, we evaluated both our sagittal and frontal plane workflows in two patient populations with neurologic damage or disease (persons post-stroke and persons with PD). We compared spatiotemporal gait parameters (step time, step length, and gait speed), lower-limb sagittal plane joint kinematics, and condition-specific, clinically relevant parameters (stroke: step time asymmetry and step length asymmetry; PD: trunk inclination) simultaneously obtained with 3D motion capture and with sagittal and frontal plane videos (data collection setup shown in Fig 4A). Note that frontal videos are limited to spatiotemporal gait parameters and that joint kinematics and trunk inclination can only be obtained from sagittal videos within our current workflows.

We present gait parameters as averaged values across four overground walking bouts each at 1) preferred and 2) fast speeds (see S2 Table for values of gait parameters). For preferred speed trials we instructed participants to walk at their preferred speed; for fast speed trials we instructed participants to walk at the fastest speed that they felt comfortable. Of the four trials at each speed, there were two trials of the participants walking away from the frontal camera (with the left side against the sagittal camera) and two trials walking toward the frontal camera (with the right side against the sagittal camera). We intend our workflows to have clinical applications and therefore present values as session-level values (i.e., the results that would be obtained as if the four walking trials were treated as a single clinical gait analysis); we report more detailed comparisons at the level of single trial averages and step-by-step comparisons in the supplement (S3 and S4 Tables).

## Testing in persons post-stroke

We then tested how well our workflows could measure gait parameters in persons post-stroke. Step time showed average differences and errors of zero and one motion capture frames (recorded at 100 Hz; 0 and 0.01 s), respectively, between motion capture and sagittal videos; and average differences and errors of two and five motion capture frames (0.02 and 0.05 s), respectively, between motion capture and frontal videos (Fig 4B and Table 1). The 95% limits of agreement spanned a narrower interval (−0.04 to 0.04 s) for sagittal videos than frontal videos (−0.09 to 0.10 s). Correlations of step time between motion capture and videos were strong (Fig 4B; all $r \geq 0.980$).

Step length showed average differences and errors of about 1 and 3 cm between motion capture and sagittal videos and average differences and errors of about −3 and 7 cm between motion capture and frontal videos (Fig 4C and Table 1). The 95% limits of agreement spanned intervals of −0.058 to 0.079 m for sagittal videos and −0.154 to 0.087 m for frontal videos. Correlations of step length between motion capture and videos were strong (Fig 4C; $r \geq 0.922$).

Gait speed showed average differences and errors of 0.02 and 0.04 m s$^{-1}$ between motion capture and sagittal videos and average differences and errors of −0.07 and 0.10 m s$^{-1}$ between

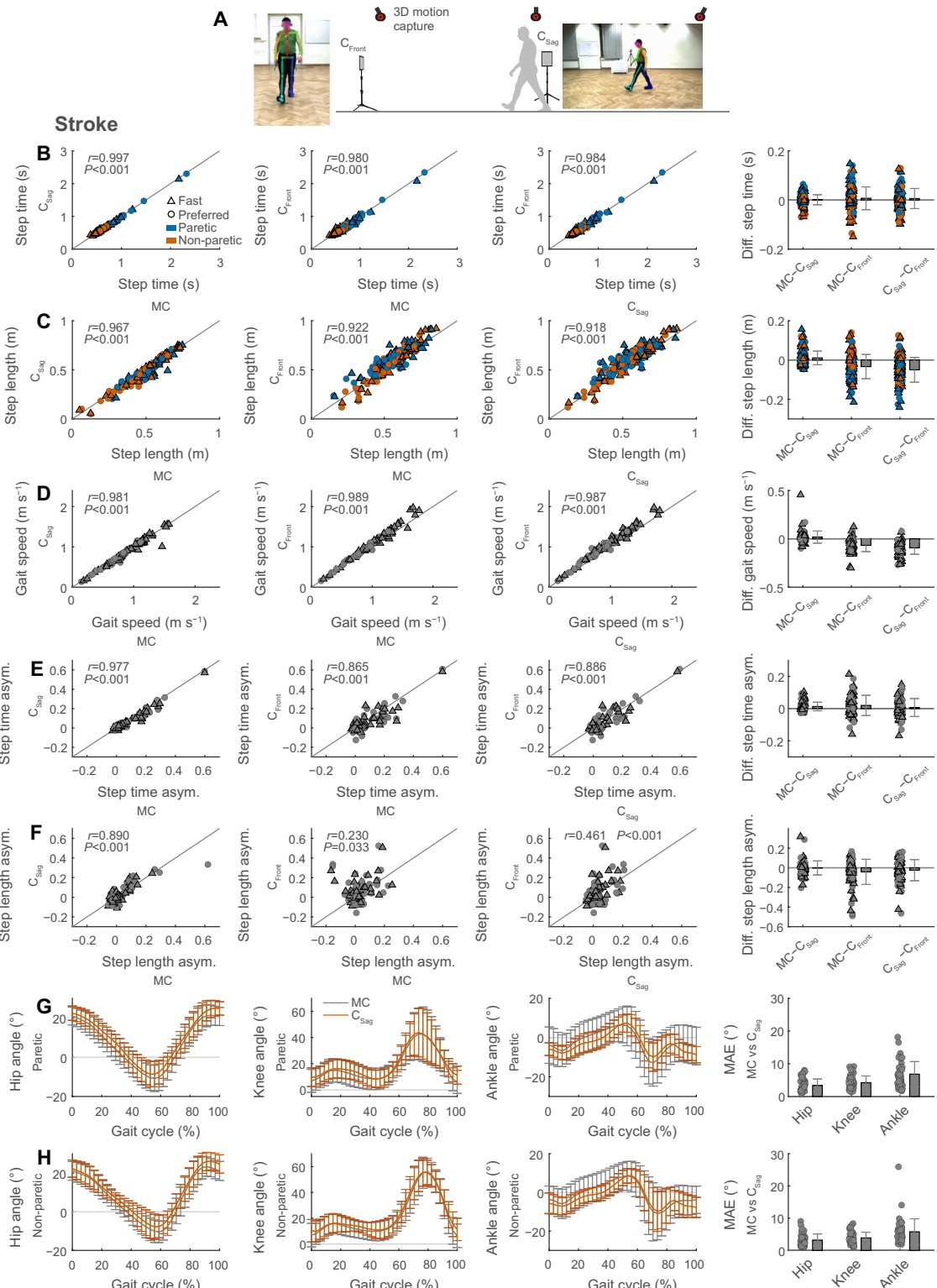

**Fig 4. Video-based gait analysis from frontal and sagittal views in persons post-stroke.** We recorded digital videos of the frontal and sagittal plane during gait trials (A). We compared spatiotemporal gait parameters (B, step time; C, step length; D, gait speed) and gait asymmetry (E, step time asymmetry; F, step length asymmetry) between the two digital videos and 3D motion capture. We also compared lower-limb joint kinematics at the hip, knee and ankle obtained with sagittal videos and motion capture for the paretic (G) and non-paretic (H) limbs (MAE, mean absolute error). Gait parameters are calculated as session-level averages of four gait trials at either preferred or fast speeds (see Table 1).

**Table 1. Comparison of video-based and motion capture measurements of spatiotemporal gait parameters in the stroke and Parkinson's disease groups [a].**

| Gait Parameter | Difference (Mean±SD) | | | Error (Mean±SD) | | | 95% Limits of Agreement | | |
|---|---|---|---|---|---|---|---|---|---|
| | $MC-C_S$ | $MC-C_F$ | $C_F-C_S$ | $\|MC-C_S\|$ | $\|MC-C_F\|$ | $\|C_S-C_F\|$ | $MC-C_S$ | $MC-C_F$ | $C_F-C_S$ |
| | | | | *Stroke* | | | | | |
| Step time (s) | 0.00±0.02 | 0.01±0.05 | 0.01±0.04 | 0.02±0.01 | 0.05±0.04 | 0.05±0.04 | −0.04; 0.04 | −0.09; 0.10 | −0.08; 0.09 |
| Step length (m) [b] | 0.010±0.035 | −0.033±0.061 | −0.050±0.063 | 0.028±0.024 | 0.072±0.037 | 0.079±0.044 | −0.058; 0.079 | −0.154; 0.087 | −0.173; 0.073 |
| Gait speed (m s$^{-1}$) [b] | 0.02±0.06 | −0.07±0.07 | −0.09±0.07 | 0.04±0.05 | 0.10±0.06 | 0.12±0.06 | −0.11; 0.14 | −0.20; 0.06 | −0.22; 0.04 |
| Step time asym. | 0.01±0.03 | 0.02±0.06 | 0.01±0.06 | 0.03±0.02 | 0.07±0.05 | 0.07±0.04 | −0.04; 0.07 | −0.10; 0.14 | −0.10; 0.12 |
| Step length asym. [b] | −0.002±0.072 | −0.042±0.127 | −0.025±0.107 | 0.050±0.053 | 0.106±0.097 | 0.100±0.073 | −0.142; 0.138 | −0.291; 0.208 | −0.235; 0.186 |
| | | | | *Parkinson's disease* | | | | | |
| Step time (s) | −0.00±0.01 | 0.01±0.02 | 0.01±0.02 | 0.01±0.00 | 0.03±0.01 | 0.03±0.01 | −0.02; 0.02 | −0.03; 0.05 | −0.03; 0.05 |
| Step length (m) [b] | −0.010±0.017 | −0.051±0.050 | −0.041±0.055 | 0.021±0.009 | 0.074±0.042 | 0.075±0.040 | −0.044; 0.023 | −0.150; 0.048 | −0.149; 0.068 |
| Gait speed (m s$^{-1}$) [b] | −0.02±0.02 | −0.12±0.08 | −0.10±0.09 | 0.03±0.02 | 0.15±0.07 | 0.15±0.06 | −0.07; 0.03 | −0.28; 0.04 | −0.27; 0.07 |
| Trunk incl. (°) [c] | −0.0±1.5 | . . . | . . . | 1.5±0.7 | . . . | . . . | −3.0; 2.9 | . . . | . . . |

MC, motion capture; $C_S$, sagittal plane camera; $C_F$, frontal plane camera

[a] Values of spatiotemporal gait parameters are calculated as session-level averages.

[b] Parameter depending on step length: comparisons of MC and $C_S$, step length calculated as distance between ankles at heel-strike; comparisons of MC and $C_F$ and of $C_S$ and $C_F$, step length calculated as distance travelled by torso between consecutive heel-strikes.

[c] Missing values because trunk inclination cannot be calculated from $C_F$.

motion capture and frontal videos (Fig 4D and Table 1). The 95% limits of agreement spanned intervals of −0.11 to 0.14 m s$^{-1}$ for sagittal videos and −0.20 to 0.06 m s$^{-1}$ for frontal videos. Correlations of gait speed between motion capture and videos were strong (Fig 4D; $r \geq 0.981$).

Step time asymmetry showed average differences and errors of 0.01 and 0.03 between motion capture and sagittal videos and average differences and errors of 0.02 and 0.07 between motion capture and frontal videos (Fig 4E and Table 1). The 95% limits of agreement spanned intervals of −0.04 to 0.07 for sagittal videos and −0.10 to 0.14 for sagittal videos. Correlations of step time asymmetry between motion capture and videos were strong (Fig 4E; all $r \geq 0.865$).

Step length asymmetry showed average differences and errors of −0.002 and 0.050 between motion capture and sagittal videos and average differences and errors of −0.042 and 0.106 between motion capture and frontal videos (Fig 4F and Table 1). The 95% limits of agreement spanned intervals of −0.142 to 0.138 for sagittal videos and −0.291 to 0.208 for frontal videos. Correlations of step length asymmetry were strong between motion capture and sagittal videos (Fig 4F; $r = 0.890$) but weak between motion capture and frontal videos (Fig 4F; $r = 0.230$).

The average mean absolute errors of lower-limb sagittal plane joint kinematics of the paretic and non-paretic limbs were 3.3°, 4.0°, and 6.3° at the hip, knee, and ankle, respectively, between motion capture and sagittal videos (Fig 4G and 4H).

## Testing in persons with Parkinson's disease

We next evaluated the performance of the video-based gait analysis in persons with PD (Fig 5A). Step time showed average differences and errors of zero and one motion capture frames (0 and 0.01 s) between motion capture and sagittal videos and average differences and errors of one and three motion capture frames (0.01 and 0.03 s) between motion capture and frontal videos (Fig 5B and Table 1). The 95% limits of agreement spanned intervals of −0.02 to 0.02 s for sagittal videos and −0.03 to 0.05 s for frontal videos. Correlations of step time between motion capture and videos were strong (Fig 5B; all $r \geq 0.961$).

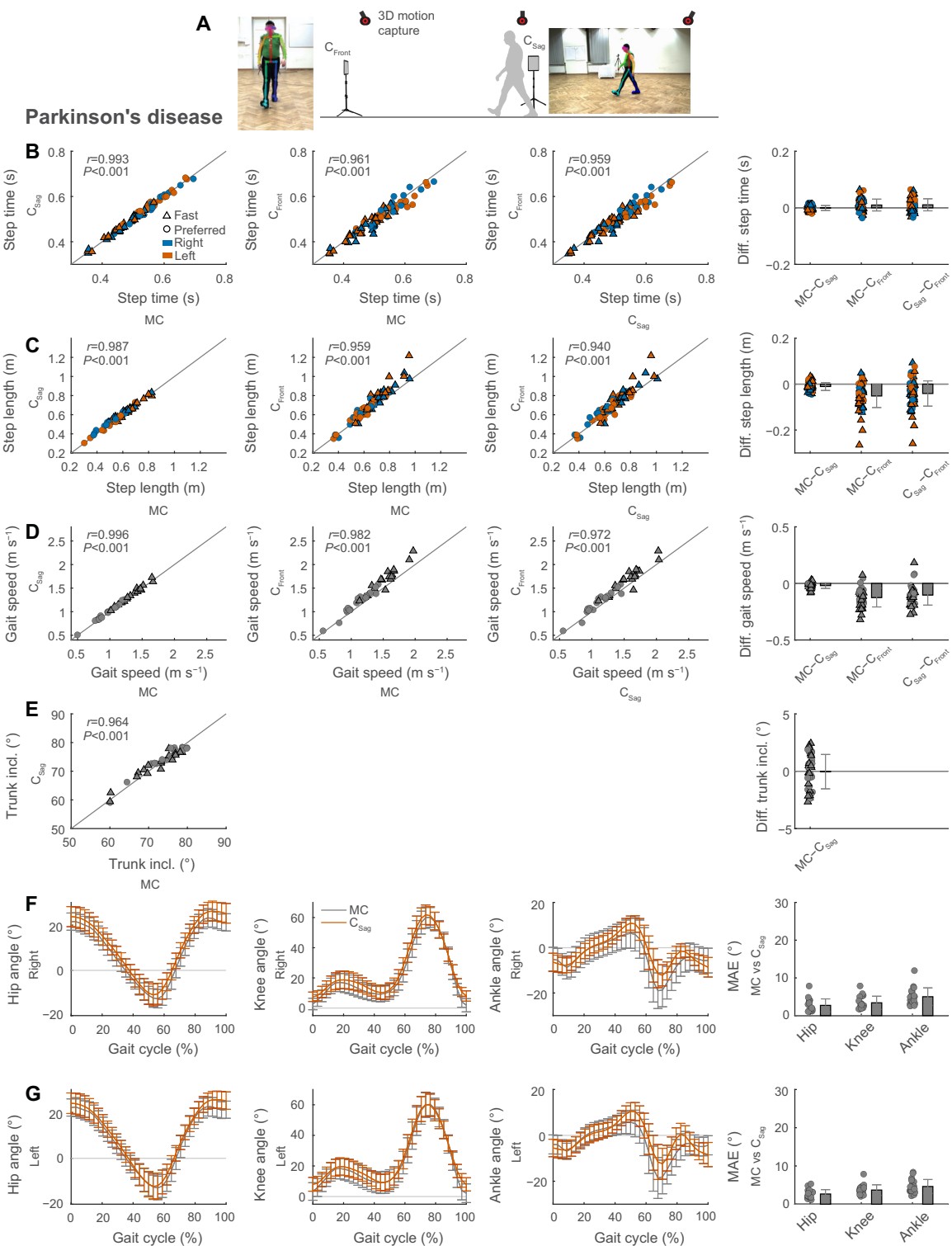

**Fig 5. Video-based gait analysis from frontal and sagittal views in persons with Parkinson's disease.** We recorded digital videos of the frontal and sagittal plane during gait trials (A). We compared spatiotemporal gait parameters (B, step time; C, step length; D, gait speed) between the two digital videos and 3D motion capture. We compared trunk inclination between sagittal plane videos and motion capture (E). We also compared lower-limb joint kinematics at the hip, knee and ankle obtained with sagittal videos and motion capture for the right (F) and non-paretic (G) limbs (MAE, mean absolute error). Gait parameters are calculated as session-level averages of four gait trials at either preferred or fast speeds (see Table 1).

Step length showed average differences and errors of about −1 and 2 cm between motion capture and sagittal videos and average differences and errors of −5 and 7 cm between motion capture and frontal videos (Fig 5C and Table 1). The 95% limits of agreement spanned intervals of −0.044 to 0.023 m for sagittal videos and −0.150 to 0.048 m for frontal videos. Correlations of step length between motion capture and videos were strong (Fig 5C; all $r \geq 0.959$).

Gait speed showed average differences and errors of −0.02 and 0.03 m s$^{-1}$ between motion capture and sagittal videos and average differences and errors of −0.12 and 0.15 m s$^{-1}$ between motion capture and frontal videos (Fig 5D and Table 1). The 95% limits of agreement spanned intervals of −0.07 to 0.03 m s$^{-1}$ for sagittal videos and −0.28 to 0.04 m s$^{-1}$ for frontal videos. Correlations of gait speed between motion capture and videos were strong (Fig 5D; all $r \geq 0.982$).

Trunk inclination showed average differences and errors of 0˚ and 1.5˚ between motion capture and sagittal videos (Fig 5E and Table 1; trunk inclination can only be extracted from sagittal videos, not frontal videos).

The average mean absolute errors of left and right lower-limb sagittal plane joint kinematics were 2.7˚, 3.5˚, and 4.8˚ at the hip, knee, and ankle, respectively, between motion capture and sagittal videos (Fig 5F and 5G).

## Measuring changes in gait that occur due to changes in gait speed

Next, to evaluate how accurately video analysis can track within-participant gait changes, we calculated the changes in spatiotemporal gait parameters that accompanied the increase in gait speed from preferred to fast speed gait trials in persons post-stroke and with PD (Fig 6A). The change in step time as a result of faster walking in persons post-stroke showed average differences and errors of zero and two motion capture frames (0 and 0.02 s) when compared between motion capture and sagittal videos and average differences and errors of zero and four motion capture frames (0 and 0.04 s) when compared between motion capture and frontal videos (Fig 6B and Table 2). The 95% limits of agreement of change in step time of post-stroke walking spanned intervals of −0.03 to 0.03 s for sagittal videos and −0.08 to 0.07 s for frontal videos.

In persons with PD, the change in step time showed average differences and error of zero and two motion capture frames (0 and 0.02 s) between motion capture and sagittal videos and average differences and errors of zero and three motion capture frames (0 and 0.03 s) between motion capture and frontal videos (Fig 6B and Table 2). The 95% limits of agreement of change in step time of PD walking spanned intervals of −0.02 to 0.02 s for sagittal videos and −0.05 to 0.04 s for frontal videos. Correlations of change in step time between motion capture and videos were strong (Fig 6B; all $r \geq 0.828$).

The change in step length as a result of faster walking in persons post-stroke showed average differences and errors of about 0 and 2 cm between motion capture and sagittal videos and average differences and errors of about −1 and 5 cm between motion capture and frontal videos (Fig 6C and Table 2). The 95% limits of agreement of change in step length of post-stroke walking spanned intervals of −0.031 to 0.037 m for sagittal videos and −0.088 to 0.075 m for frontal videos.

Change in step length in persons with PD showed average differences and errors of about 0 and 2 cm between motion capture and sagittal videos and average differences and errors of about −3 and 7 cm between motion capture and frontal videos (Fig 6C and Table 2). The 95% limits of agreement of change in step length of PD walking spanned intervals of −0.022 to

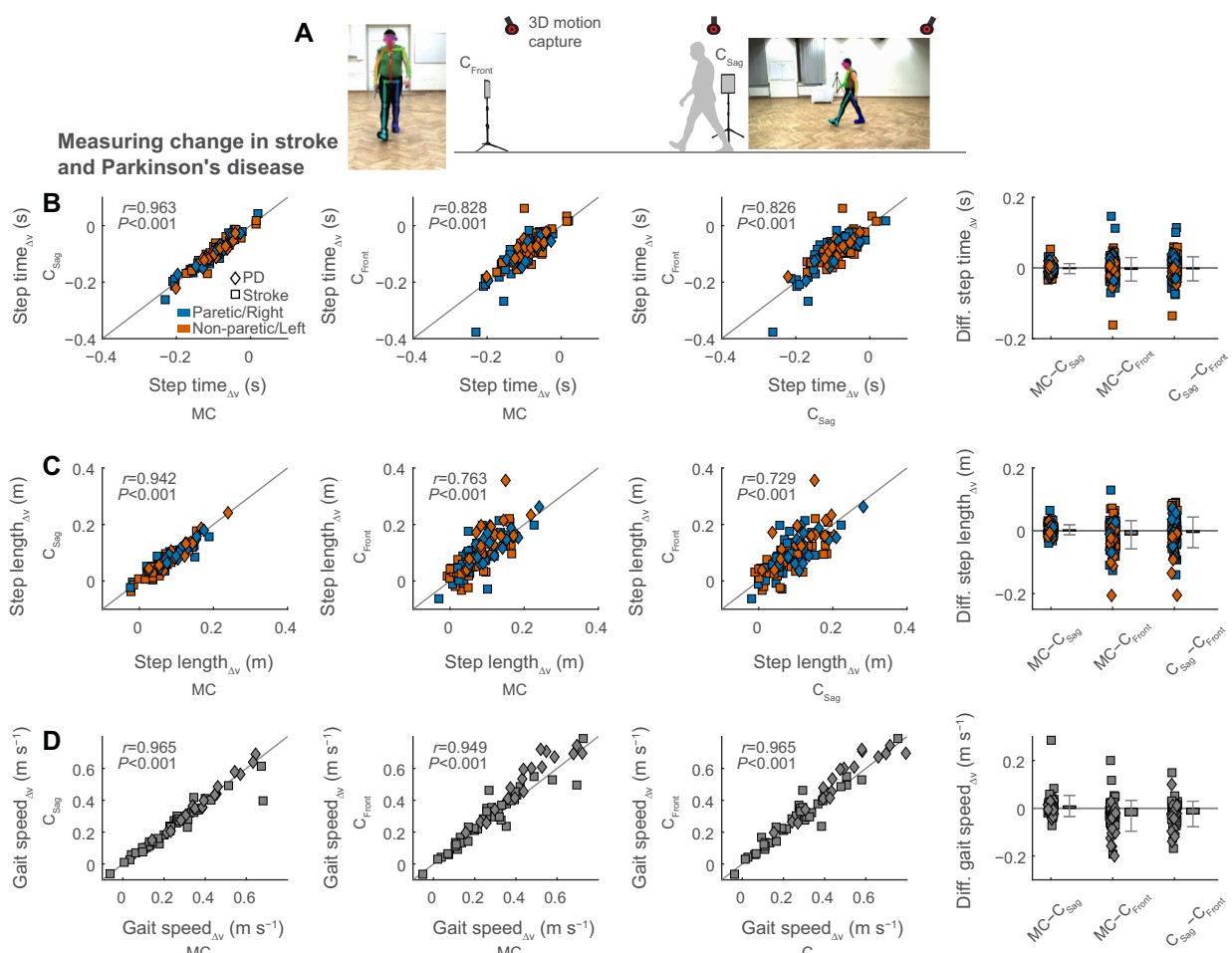

**Fig 6. Measuring changes in gait that occur due to changes in gait speed in persons post-stroke and persons with Parkinson's disease.** We recorded digital videos of the frontal and sagittal plane during gait trials at preferred and fast speeds (A). We compared spatiotemporal gait parameters (B, step time; C, step length; D, gait speed) between the two digital videos and 3D motion capture. Subscripts Δv of gait parameters denote changes in the gait parameter due to speed-increases from preferred to fast speed walking trials. We calculated gait parameters as the difference between the session-level averages of preferred and fast speed trials (see Table 2).

0.028 m for sagittal videos and −0.122 to 0.070 m for frontal videos. Correlations of change in step length between motion capture and videos were strong (Fig 6C; all $r \geq 0.763$).

The change in gait speed from preferred to fast speed gait trials in persons post-stroke showed average differences and errors of 0.01 and 0.04 m s$^{-1}$ between motion capture and sagittal videos and average differences and errors of −0.02 and 0.06 m s$^{-1}$ between motion capture and frontal videos (Fig 6D and Table 2). The 95% limits of agreement of change in gait speed of post-stroke walking spanned intervals of −0.09 to 0.11 m s$^{-1}$ for sagittal videos and −0.14 to 0.11 m s$^{-1}$ for frontal videos.

Finally, in persons with PD, measured change in gait speed showed average differences and errors of 0 and 0.03 m s$^{-1}$ between motion capture and sagittal videos and average differences and errors of −0.07 and 0.11 m s$^{-1}$ between motion capture and frontal videos (Fig 6D and Table 2). The 95% limits of agreement of change in gait speed of PD walking spanned intervals of −0.04 to 0.04 m s$^{-1}$ for sagittal videos and −0.19 to 0.06 m s$^{-1}$ for

**Table 2. Comparison of video-based and motion capture measurements of speed-related changes of spatiotemporal gait parameters of stroke and PD groups [a].**

| Gait Parameter | Difference (Mean±SD) | | | Error (Mean±SD) | | | 95% Limits of Agreement | | |
|---|---|---|---|---|---|---|---|---|---|
| | MC−C$_S$ | MC−C$_F$ | C$_F$−C$_S$ | \|MC−C$_S$\| | \|MC−C$_F$\| | \|C$_S$−C$_F$\| | MC−C$_S$ | MC−C$_F$ | C$_F$−C$_S$ |
| *Stroke* | | | | | | | | | |
| Step time (s) | −0.00±0.01 | −0.00±0.04 | −0.00±0.04 | 0.02±0.01 | 0.04±0.03 | 0.05±0.03 | −0.03; 0.03 | −0.08; 0.07 | −0.07; 0.07 |
| Step length (m) [b] | 0.003±0.017 | −0.007±0.042 | −0.001±0.044 | 0.021±0.012 | 0.054±0.027 | 0.058±0.029 | −0.031; 0.037 | −0.088; 0.075 | −0.087; 0.085 |
| Gait speed (m s$^{-1}$) [b] | 0.01±0.05 | −0.02±0.06 | −0.02±0.05 | 0.04±0.04 | 0.06±0.05 | 0.06±0.04 | −0.09; 0.11 | −0.14; 0.11 | −0.12; 0.08 |
| *Parkinson's disease* | | | | | | | | | |
| Step time (s) | −0.00±0.01 | −0.00±0.02 | −0.00±0.02 | 0.02±0.01 | 0.03±0.02 | 0.04±0.02 | −0.02; 0.02 | −0.05; 0.04 | −0.05; 0.04 |
| Step length (m) [b] | 0.003±0.013 | −0.026±0.049 | −0.015±0.056 | 0.019±0.007 | 0.067±0.035 | 0.073±0.039 | −0.022; 0.028 | −0.122; 0.070 | −0.125; 0.094 |
| Gait speed (m s$^{-1}$) [b] | 0.00±0.02 | −0.07±0.06 | −0.04±0.06 | 0.03±0.01 | 0.11±0.06 | 0.10±0.05 | −0.04; 0.04 | −0.19; 0.06 | −0.16; 0.08 |

MC, motion capture; C$_S$, sagittal plane camera; C$_F$, frontal plane camera

[a] Speed-changes are differences between preferred and fast speed walking trials; gait parameters are calculated as session-level averages.

[b] Parameter depending on step length: comparisons of MC and C$_S$, step length calculated as distance between ankles at heel-strike; comparisons of MC and C$_F$ and of C$_S$ and C$_F$, step length calculated as distance travelled by torso between consecutive heel-strikes.

frontal videos. Correlations of change in gait speed between motion capture and videos were strong (Fig 6D; all $r \geq 0.949$).

## Factors that affect accuracy of the frontal video-based gait analysis workflow

We noted that step length errors were occasionally large when calculated from frontal videos (up to nearly 30% of the average step length). We have previously described factors such as the position of the person relative to the camera that influence step length errors when calculated from sagittal videos [11]. Similarly, we wanted to identify and understand factors that influence step length errors from videos recorded in the frontal plane.

First, we considered that greater depth of the person relative to the frontal plane camera may lead to less precise step length estimates (S1 Fig). We partitioned the analysis of step length errors into videos from the frontal plane where the person walks away from the camera or toward the camera because OpenPose may track keypoints differently when viewing the front of the person (when walking toward) or the back of the person (when walking away). We found that step length errors increased with greater depth from the camera, so that the person's size appeared smaller in the image. Step length errors were more affected by depth when the person walked away from the camera compared to walking toward the camera: from average step length errors of about 7 cm nearest the camera (beginning of trial when the person walks away from the camera; end of the trial when the person walks toward the camera), average errors increased up to about 16 cm when the person walked away, with a more modest increase of up to 11 cm when the person walked toward the camera. This suggests that precision may decrease as the person appears smaller, likely due to less precise keypoint tracking by OpenPose.

We also considered whether a scaling effect influenced step length errors so that longer steps had greater errors. We found that step length errors were not influenced by the magnitude of step length (S2 Fig).

We noted time-lags in the gait cycle detection of the frontal videos relative to motion capture that could have influenced step length errors (this analysis could only be performed for the unimpaired participant dataset, in which motion capture and video recordings were synchronized). The timing of gait cycle detection differed depending on walking direction: when

the person walked away from the camera, gait cycle timings were, on average, four motion capture frames (~0.04 s) *before* the timing detected from motion capture, and 15 motion capture frames (~0.15 s) *after* motion capture when the person walked toward the camera (S3 Fig, panel A). Using gait event timings from motion capture to calculate step lengths from frontal videos, there was a statistical difference in step length errors when the person walked away from the camera ($P = 0.024$), but not when the person walked toward the camera ($P = 0.501$; S3 Fig, panel B). The average step length error decreased from about 2 to 1 cm in the unimpaired participant dataset when using gait event timing from the motion capture data in the videos where the person walked away from the camera.

Last, we considered that walking direction relative to the frontal plane camera may have influenced the accuracy of gait parameters. In the unimpaired participant dataset, in which two frontal plane cameras simultaneously captured the same walking trial from different vantage points (see Fig 3A), we noted a minor overestimation of gait speed by an average of $0.04 \text{ m s}^{-1}$ from the camera that the person walked away from compared to the camera that the person walked toward (S1 Table). We observed similar, albeit exaggerated, trends in the stroke and PD datasets. When comparing the average gait speed differences between motion capture and the frontal plane camera, gait speed was overestimated by 0.13 and $0.21 \text{ m s}^{-1}$ for stroke and PD, respectively, when the person walked toward the frontal plane camera; the overestimation was only minor at 0.01 and $0.03 \text{ m s}^{-1}$ for stroke and PD, respectively, when the person walked toward the camera (S3 Table). The overestimation of gait speed was accompanied by greater errors when comparing the frontal camera to motion capture: average errors were 0.14 and $0.23 \text{ m s}^{-1}$ for stroke and PD, respectively, when the person walked away from the camera; errors were only 0.06 and $0.08 \text{ m s}^{-1}$ when the person walked toward the camera (S3 Table).

The trends of overestimation and greater errors from frontal plane recordings where the person walked away from the camera were mirrored in the results of step length: there were greater overestimations and errors of step length when the person walked away from the camera (average overestimations of 0.056 and 0.082 m and errors of 0.084 and 0.092 m for stroke and PD, respectively) compared to when the person walked toward the camera (S3 Table; average overestimations of 0.013 and 0.021 m and errors of 0.062 and 0.055 m for stroke and PD, respectively). This suggests that spatial gait parameters obtained from a frontal plane camera are influenced by walking direction and that the greatest precision was obtained when the person walked toward the camera. Furthermore, this also suggests that the accuracy of gait parameters presented here, when calculated as session-level averages, can be improved if using only gait trials with the same walking direction.

## Discussion

In this study, we demonstrated a new approach for performing clinical gait analyses using simple videos recorded using low-cost devices and a workflow that leverages a freely available pose estimation algorithm (OpenPose) for video-based movement tracking. We showed that this novel approach can perform accurate gait analyses 1) from videos recorded from multiple perspectives (e.g., frontal or sagittal viewpoints), 2) across a diverse range of persons with and without gait impairment, 3) that capture clinically relevant and condition-specific aspects of gait, and 4) that measure within-participant changes in gait as a result of changes in walking speed. These findings demonstrate the versatility and accessibility of video-based gait analysis and have significant potential for clinical applications.

Interest in video-based, markerless gait analysis has accelerated rapidly. Previous studies have used various approaches to move quantitative clinical gait analysis outside of the laboratory or research center and directly into the home or clinic [5,6,13–15,17]. Here, we aimed to

develop a single approach that addressed several outstanding needs, including the needs to accommodate multiple different types of environments/viewing perspectives, use of datasets in multiple clinical populations with gait impairment, measurement of both spatiotemporal gait parameters and lower extremity two-dimensional kinematics, and measurement of within-participant changes in gait. It is also notable that we achieved accurate results using multiple different video recording devices with different sampling rates. By comparing our results against gold standard motion capture measurements, we provide data about the accuracy of all findings with respect to the current state-of-the-art.

Our findings also enable us to progress toward development of a series of best practices for video-based clinical gait analysis. Unsurprisingly, we found that video-based gait analyses generated from videos recorded using a sagittal viewpoint generally led to stronger correlations with motion capture data and lower error when compared to videos recorded from frontal viewpoints. This was particularly evident in gait parameters that require especially high levels of precision (e.g., step length asymmetry in persons post-stroke). Similar to our previous work [11], we also found that video-based measurements of ankle kinematics were generally less accurate (relative to motion capture) than measurements of hip or knee kinematics in persons with or without gait impairment. Therefore, when using the current iteration of our workflow, a user is likely to obtain best results by recording a sagittal video (if possible) and targeting measurement of spatiotemporal gait parameters and more proximal lower limb kinematics. We emphasize that our single-camera, video-based approach is not intended to reach marker-based motion capture levels of accuracy that other multi-camera approaches may target [6,18,19] or that may be required by various scientific disciplines (e.g., biomechanics, human motor control), but rather offers clinicians and other end-users access to a reasonably accurate approach for clinical gait analysis that requires minimal time and only a single video recording device.

It is informative to consider the accuracy of our workflow relative to reported test-retest minimal detectable change or minimal clinical important difference values of the population of interest. For example, a meaningful change in gait speed is often reported as 0.10 m s$^{-1}$ [20], but may vary from 0.05 up to 0.30 m s$^{-1}$ depending on the population studied [21–32]. The average errors of our video-based measurements relative to motion capture generally fall within these margins, suggesting that gait speed is likely to be reliably measured in many populations (e.g., older adults, post-stroke, PD, following hip fracture, cerebral palsy, multiple sclerosis) using our workflow. Minimal detectable changes in gait kinematics may also be dependent on the population of interest, with estimates ranging from about 4˚ to 11˚ of lower-limb sagittal plane kinematics [26,28,33–36]. Average errors of sagittal plane hip and knee kinematics in our study were less than 4˚, while errors at the ankle were up to 6.8˚, suggesting that hip and knee kinematics from our workflow can be accurately tracked while continued improvement in measurement of ankle angles is needed.

There remain additional significant hurdles to widespread implementation of video-based clinical gait analyses. There is a crucial unmet need for improved ease of use, as the user currently must have access to specific computing hardware (i.e., pose estimation is most efficient when using a graphics processing unit (GPU)), download all relevant software, record the videos, and manually process each video through the workflow. This generates an output that is contained within the software. This process is not well-suited for users without some level of technical expertise; there is an important need for new technologies that can streamline these steps and remove much of the technical know-how and burden of manual processing. Furthermore, there is a need for validation in additional adult and pediatric clinical populations, as previous work has shown that existing pose estimation algorithms have difficulty with tracking patient populations with anatomical structures that likely differ significantly from the images

used to train the algorithms [13]. Thirteen of the participants with stroke used a cane; we did not observe instances where OpenPose mistakenly identified the cane as a limb. Lastly, it is likely that accuracy will continue to improve in the future as both computer vision algorithms and methods for data post-processing continue to advance. In this study, we used a pre-trained network [8], while a different network that was trained to be specific to both gait and clinical condition may further improve accuracy (the challenges of existing pre-trained networks for human pose estimation in movement science have been well-documented [37]).

In this study, we developed and tested a novel approach for video-based clinical gait analysis. We showed that this approach accommodates multiple viewing perspectives, provides accurate and clinically relevant gait analyses (as compared to 3D motion capture) across multiple participant populations with and without gait impairment, and tracks within-participant changes in gait that are relevant to rehabilitation and recovery outcomes. All software needed to perform these analyses is freely available at https://github.com/janstenum/GaitAnalysis-PoseEstimation/tree/Multiple-Perspectives, where we also provide a series of detailed instructions to assist the user. There is an urgent need to begin to move these emerging technologies with potential for significant clinical applications toward more user-friendly solutions.

## Materials and methods

### Participants

We recruited 44 individuals post-stroke (15 female, 29 male; age 61±11 years (mean±SD); body mass 90±23 kg; height 1.73±0.11 m) and 19 individuals with PD (6 female, 13 male; age 67±7 years; body mass 77±14 kg; height 1.71±0.09 m) to participate in the study; all participants were capable of walking independently with or without an assistive device. All participants gave written informed consent before enrolling in the study in accordance with the protocol approved by The Johns Hopkins School of Medicine Institutional Review Board (Protocol IRB00255175). Additionally, we used a publicly available dataset [38] of overground walking sequences from 32 unimpaired participants (10 women, 22 men) made available at http://bytom.pja.edu.pl/projekty/hm-gpjatk. The dataset included synchronized 3D motion capture files and digital video recordings of the walking sequences. The publicly available dataset does not contain identifiable participant information and faces have been blurred in the video recordings. Our analysis of the publicly available videos was deemed exempt by The Johns Hopkins University School of Medicine Institutional Review Board.

### Protocol and data collection

Participants visited our laboratory for one day of testing. They first performed ten-meter walk tests at their preferred speed and the fastest speed at which they felt comfortable walking. Participants then performed eight overground walking trials (four trials at each preferred and fast speeds) across a walkway of 4.83 m.

We mounted two commercially available tablets (Samsung Galaxy Tab A7) on tripods positioned to capture frontal ($C_{Front}$) and sagittal ($C_{Sag}$) plane views of the overground walking trials (video recordings occurred at a 30-Hz sampling rate; see Fig 1 for overview). Of the eight total walking trials, the participant walked away from the frontal plane camera with the left side turned to the sagittal plane camera during four of the trials; during the other four trials, the participant walked toward the frontal plane camera with the right side turned to the sagittal plane camera. Tablet cameras obtained videos with 1920 × 1080 pixel resolution. The frontal-view tablet was positioned 1.52 m behind the start/end of the walkway and the sagittal-view tablet was positioned 3.89 m to the side of the midpoint of the walkway. The tablet positions were chosen to achieve the longest walkway in which the person remained visible to both frontal and sagittal

tablets, given the space restrictions of the laboratory. The frontal- and sagittal-view tablets were rotated to capture portrait and landscape views, respectively. The height of the frontal-view camera was set so that the entire participant remained visible when they were nearest the camera (about 0.85 m). The height of the sagittal-view camera was about 1.18 m so that the participant appeared in the middle of the image as they travelled across the walkway.

We simultaneously recorded walking trials using ten cameras (Vicon Vero, Denver, CO, USA) as part of a marker-based, 3D motion capture system at 100 Hz. We placed reflective markers on the seventh cervical vertebrae (C7), tenth thoracic vertebrae, jugular notch, xiphoid process, and bilaterally over the second and fifth metatarsal heads, calcaneus, medial and lateral malleoli, shank, medial and lateral femoral epicondyles, thigh, greater trochanter, iliac crest, and anterior and posterior superior iliac spines (ASIS and PSIS, respectively).

In the previously published dataset of unimpaired adults without gait impairment, we used a subset of the data (sequences labelled *s1*) that consisted of a single walking bout of approximately 5 m that included gait initiation and termination. We excluded data for one participant because the data belonged to another subset with diagonal walking sequences. We used data from two digital cameras (Basler Pilot piA1900-32gc, Ahrensburg, Germany) that simultaneously recorded frontal plane views of the person walking away from one camera and toward the other camera (see Fig 3A for overview). The digital cameras obtained videos with $960 \times 540$ pixel resolution captured at 25 Hz. The average distance from the starting position of the participants to the cameras were 2.50 and 7.28 m for the camera that recorded the participant walking away and toward, respectively. Cameras were mounted on tripods and the height was about 1.3 m. Motion capture cameras (Vicon MX-T40, Denver, CO, USA) recorded 3D marker positions at 100 Hz. Markers were placed on the seventh cervical vertebrae, tenth thoracic vertebrae (T10), manubrium, sternum, right upper back and bilaterally on the front and back of the head, shoulder, upper arm, elbow, forearm, wrist (at radius and ulna), middle finger, ASIS, PSIS, thigh, knee, shank, ankle, heel, and toe.

## Data processing and analysis

Motion capture data from the participants with stroke or PD were smoothed using a zero-lag $4^{th}$ order low-pass Butterworth filter with a cutoff frequency of 7 Hz. The motion capture data from the participants without gait impairment in the publicly available dataset had already been smoothed. We identified left and right heel-strikes and toe-offs as the positive and negative peaks, respectively, of the anterior-posterior left or right ankle markers relative to the torso [39].

All digital video data were processed in two steps: 1) using OpenPose to automatically detect and label two-dimensional coordinates of various anatomical keypoints, 2) post-processing in MATLAB using custom-written code. The OpenPose analysis was similar for all video data, whereas we divided the post-processing workflows into two separate pipelines for videos capturing frontal or sagittal plane views.

1. OpenPose Analysis

   a. We ran the OpenPose demo over sequences of the video recordings that contained each walking bout. We have previously used a cloud-based service to run OpenPose with remote access to GPUs. Here we used a local computer with a GPU (NVIDIA GeForce RTX 3080) so that videos containing identifiable participant information were not shared with any third-party services.

   b. Videos were analyzed in OpenPose using the BODY_25 keypoint model that tracks the following 25 keypoints: nose, neck, mid-hip and bilateral keypoints at the eyes, ears, shoulders, elbows, wrists, hips, knees, ankles, heels, halluces, and fifth toes.

c. The output of the OpenPose analysis yielded: 1) JSON files for every video frame containing pixel coordinates of each keypoint detected in the frame, and 2) a new video file in which a stick figure that represents the detected keypoints is overlaid on the original video recording.

2. MATLAB Post-processing

We created custom-written MATLAB code to process the JSON files that were output from the OpenPose analysis (https://github.com/janstenum/GaitAnalysis-PoseEstimation/tree/Multiple-Perspectives). As an initial step, we checked whether multiple persons had been detected by OpenPose in the video (this can be the case when multiple people are visible or when OpenPose incorrectly detects keypoints in inanimate objects). Note that OpenPose has an optional flag to track only a single person; however, we did not use this option to avoid scenarios where the participant had not been tracked in favor of other persons (e.g., the experimenter). If multiple persons were detected, three MATLAB scripts were called that 1) required user input to identify the participant in a single frame of the video, 2) automatically identified the participant throughout the video and 3) allowed the user to visually inspect that the participant had been identified and correct any errors. Following the person-identification step, MATLAB workflows were different depending on whether the camera captured a frontal or sagittal plane view of the walking trial. We describe each workflow below.

a. Frontal plane videos

i. We changed the pixel coordinate system so that the positive vertical was directed upward and that positive horizontal was directed toward the participant's left side.

ii. We visually inspected and corrected errors in left-right identification of the limbs. In all, 362 (less than 1% of the 131,519 frames in total) frontal video frames were corrected.

iii. We gap-filled keypoint trajectories using linear interpolation for gaps spanning to up 0.12 s.

iv. We identified events of left and right gait cycles by local maxima and minima of the vertical distance between the left and right ankle keypoints. Gait events on the left limb were detected at positive peaks and gait events on the right limb were detected at negative peaks in trials where the participants walked away from the frontal plane camera; and vice versa in trials where the participants walked toward the camera. In order to unify the nomenclature of gait events across motion capture data and sagittal and frontal plane video data, we refer to the gait events of the frontal plane analysis as heel-strikes.

v. Last, we calculated a time-series of depth-change of the torso relative to the initial starting depth. We used the following equation to calculate depth-change ($\Delta d_i$):

$$\Delta d_i = \frac{d_{\text{Ref}}}{s_{\text{Ratio}}} - d_{\text{Ref}}, \qquad (\text{Eq1})$$

where $d_{\text{Ref}}$ is the initial reference depth of the person relative to the frontal camera position and $s_{\text{Ratio}}$ is the ratio of the pixel size of the person relative to the pixel size of the torso at the initial reference depth. Eq 1 is derived from trigonometric relations between the actual size of the person and the pixel size of the person as they appear on the image plane of the camera (see Fig 2A and 2B for an overview). We assume a fixed position of a pinhole camera with no lens distortion. We know the following relation when the

person is at an initial reference depth from the camera:

$$\frac{s_{\text{Ref}}}{f} = \frac{s}{d_{\text{Ref}}}, \tag{Eq2}$$

where $f$ is the focal length, $s_{\text{Ref}}$ is the pixel size of the person at the reference distance and $s$ is the actual size of the person. With a depth-change $\Delta d_i$ we obtain the following relationship:

$$\frac{s_i}{f} = \frac{s}{d_{\text{Ref}} + \Delta d_i}, \tag{Eq3}$$

where $s_i$ is the pixel size of the person as they appear with a depth-change. From Eqs 2 and 3, we obtain:

$$s_{\text{Ratio}} = \frac{s_i}{s_{\text{Ref}}} = \frac{d_{\text{Ref}}}{d_{\text{Ref}} + \Delta d_i}. \tag{Eq4}$$

Using Eq 4 we obtain the expression in Eq 1. From Eq 1 we can estimate depth changes using only information about the reference depth of the person and the pixel size of the person. We validated this approach in Fig 2C by comparing the predicted value of $s_{\text{Ratio}}$ based on Eq 4 (with a reference depth of 4.88 m) with values of $s_{\text{Ratio}}$ found by manually tracking the pixel size of images of a person standing at depth-changes up to 18.29 m. The predicted relationship closely tracks the manually annotated pixel sizes in Fig 2C, suggesting that Eq 1 can be used to accurately calculate depth-changes in the frontal plane.

Next, we considered methodological factors that may affect accuracy of the calculated depth-changes. We chose to track the size of the torso because there are only minor rotations in the transverse plane during the gait cycle, which ensures a consistent perspective during a gait trial [40]. Torso size can be represented by 1) torso height (vertical distance between neck and midhip keypoints), 2) shoulder width (horizontal distance between left and right shoulder keypoints) and 3) the torso area (calculated as the square root of the product of torso height and shoulder width to ensure that size scales appropriately with Eq 1). We evaluated the best tracking and smoothing method from the combination that yielded the lowest step length error and SD of step length differences between motion capture and frontal plane videos (See S4 Fig). Based on the evaluation, we chose to track torso size and low-pass filter size ratio using a cutoff frequency at 0.4 Hz.

b. Sagittal plane videos

 i. We changed the pixel coordinate system so that positive vertical was direction upward and positive horizontal was the direction of travel.

 ii. We visually inspected and corrected errors in left-right identification of the limbs. In all, 5,369 (about 3.5% of the 153,669 frames in total) of sagittal video frames were corrected.

 iii. We gap-filled keypoint trajectories using linear interpolation for gaps spanning up to 0.12 s.

 iv. We smoothed trajectories using a zero-lag 4$^{\text{th}}$ order low-pass Butterworth filter with a cutoff frequency at 5 Hz.

v. We calculated a scaling factor to dimensionalize pixel distance. The scaling factor was as a ratio of a known distance in the line of progression relative to the pixel distance. We used the distance between strips of tape on the walkway.

vi. We identified left and right heel-strikes and toe-offs as the positive and negative peaks, respectively, of the horizontal trajectories of the left or right ankle keypoints relative to the mid-hip keypoint.

We cross-referenced gait events that had independently been identified in motion capture data and sagittal or frontal plane video data to ensure that all gait parameters were obtained based on the same gait cycles.

We calculated the following spatiotemporal gait parameters:

- Step time: duration between consecutive bilateral heel-strikes.

- Step length (we used two methods to calculate step lengths): 1) as the horizontal distance between ankle markers or keypoints at instants of heel-strike and 2) as the distance travelled by the torso between consecutive bilateral heel-strikes. We used the distance travelled by the torso because the distances between the ankles at a heel-strike instant cannot be obtained from frontal plane videos. When comparing step lengths between motion capture and sagittal plane videos, we used the distance between the ankles; all step length comparisons with frontal plane data used the distance travelled by the torso. Step length methods were highly correlated ($r = 0.938$) with an average difference of $-0.069$ m, suggesting that the distance travelled by the torso was about 7 cm longer than the distance between the ankles (S5 Fig).

- Gait speed: step length divided by step time.

In stroke and PD data, we calculated paretic/non-paretic or left/right step times and step lengths, respectively. Paretic/left step time is the duration from non-paretic/right heel-strike until paretic/left heel-strike; vice versa for non-paretic/right step times. Paretic/left step length, calculated as the distance between the ankles, is the distance at paretic/left heel-strike; vice versa for non-paretic/right step lengths. Paretic/left step length, calculated as the distance travelled by the torso, is the distance travelled from non-paretic/right heel-strike to paretic/left heel-strike; vice versa for non-paretic/right step lengths.

We calculated the changes in spatiotemporal gait parameters that accompany speed-changes (i.e., shorter step times, longer step lengths, and faster gait speeds) from the preferred and fast speed trials in the stroke and PD data. This allowed us to test how well gait changes can be tracked using video recordings.

There are several commonly observed, clinically relevant gait impairments in stroke (e.g., gait asymmetry [41]) and PD (e.g., stooped posture [42])–thus, for each population we calculated condition-specific gait parameters. We calculated step time asymmetry and step length asymmetry (difference between steps divided by sum of steps) in stroke gait and trunk inclination in PD gait. Trunk inclination was calculated as the angle relative to vertical between the mid-hip and neck keypoints at heel-strikes in the sagittal plane videos and the angle between the C7 and right PSIS markers at heel-strikes in the motion capture data. During initial comparisons we found an offset (mean±SD 12.0˚±1.5˚) between motion capture and sagittal plane video data; we subtracted a fixed offset of 12˚ from trunk inclination in the sagittal plane video data in order to create a better numeric comparison with the motion capture data. The offset is a consequence of the fact that video-based keypoints and markers track similar anatomical regions, but do not track the *exact* same anatomical locations [37,43].

We calculated sagittal plane lower limb joint kinematics at the hip, knee, and ankle using two-dimensional coordinates from the motion capture data and the sagittal plane video data.

We used markers at the greater trochanter and lateral femoral epicondyles and keypoints at the hip and knee to calculate hip angles; markers at the greater trochanter, lateral femoral epicondyles and lateral malleoli and keypoints at the hip, knee, and ankle to calculate knee angles; markers at the lateral femoral epicondyles, lateral malleoli, and $5^{th}$ metatarsal and keypoints at the knee, ankle, and hallux to calculate ankle angles.

From our stroke and PD datasets, we compared gait parameters at three levels of comparisons: at the step level calculating parameters for individual steps, as averages across single gait trials, and at the session level calculated as averages across several gait trials. In total there were 2,684 individual gait cycles (1,790 for stroke, 709 for PD and 185 for unimpaired), 527 gait trials (352 for stroke, 144 for PD and 31 for unimpaired) and 124 session level averages (88 for stroke and 36 for PD). We present session level gait parameters for stroke and PD and trial level for unimpaired data in the main text of the manuscript; we show results at the trial and step level in the S3 and S4 Tables.

In the stroke and PD datasets, we compared gait parameters obtained during trials that were simultaneously recorded by motion capture, sagittal plane videos, and frontal plane videos (see Fig 1 for overview). Note that some parameters (joint kinematics and trunk inclination) can only be obtained with motion capture data and sagittal plane videos.

In the dataset with unimpaired participants, we compared spatiotemporal gait parameters obtained during trials that were simultaneously captured with motion capture data and with two frontal cameras positioned to capture the participant walking away from one camera and toward the other camera (see Fig 3A for overview).

## Statistical analyses

We compared gait parameters obtained with motion capture and video by calculating differences, errors (absolute differences) and 95% limits of agreement (mean differences $\pm 1.96 \times SD$). We assessed correlations by calculating Pearson correlation coefficients.

## Supporting information

**S1 Fig. Step length errors and differences of frontal plane workflow relative to person's distance to camera.**
(PDF)

**S2 Fig. Step length errors and differences of frontal plane workflow relative to magnitude of step length.**
(PDF)

**S3 Fig. Influence of gait event timings on step length errors when using frontal plane workflow.**
(PDF)

**S4 Fig. Evaluation of tracking methods and smoothing using frontal plane workflow.**
(PDF)

**S5 Fig. Comparison of two methods to calculate step length.**
(PDF)

**S1 Table. Comparison of spatiotemporal gait parameters of the unimpaired group.**
(PDF)

**S2 Table. Spatiotemporal gait parameters for stroke and PD groups.**
(PDF)

**S3 Table. Comparison of spatiotemporal gait parameters of stroke and PD groups calculated as trial averages.**
(PDF)

**S4 Table. Comparison of spatiotemporal gait parameters of stroke and PD groups calculated for individual steps.**
(PDF)

## Author Contributions

**Conceptualization:** Jan Stenum, Ryan T. Roemmich.

**Data curation:** Jan Stenum, Melody M. Hsu.

**Formal analysis:** Jan Stenum.

**Funding acquisition:** Ryan T. Roemmich.

**Investigation:** Jan Stenum.

**Methodology:** Jan Stenum, Ryan T. Roemmich.

**Project administration:** Ryan T. Roemmich.

**Resources:** Alexander Y. Pantelyat.

**Software:** Jan Stenum.

**Supervision:** Ryan T. Roemmich.

**Visualization:** Jan Stenum.

**Writing – original draft:** Jan Stenum, Ryan T. Roemmich.

**Writing – review & editing:** Jan Stenum, Melody M. Hsu, Alexander Y. Pantelyat, Ryan T. Roemmich.

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
