## [Decision Letter · Decision Letter 0]

5 Jul 2023

PDIG-D-23-00173

Clinical gait analysis using video-based pose estimation: multiple perspectives, clinical populations, and measuring change

PLOS Digital Health

Dear Dr. Roemmich,

Thank you for submitting your manuscript to PLOS Digital Health. After careful consideration, we feel that it has merit but does not fully meet PLOS Digital Health's publication criteria as it currently stands. Therefore, we invite you to submit a revised version of the manuscript that addresses the points raised during the review process.

Please submit your revised manuscript within 60 days Sep 03 2023 11:59PM. If you will need more time than this to complete your revisions, please reply to this message or contact the journal office at digitalhealth@plos.org. Please include the following items when submitting your revised manuscript:

We look forward to receiving your revised manuscript.

Kind regards,

Mengling Feng

Academic Editor

PLOS Digital Health

Journal Requirements:

1. We ask that a manuscript source file is provided at Revision. Please upload your manuscript file as a .doc, .docx, .rtf or .tex.

Additional Editor Comments (if provided):

Please study the comments from our reviewers carefully and improve your manuscript accordingly.

Reviewers' comments:

Reviewer's Responses to Questions

**Comments to the Author**

1. Does this manuscript meet PLOS Digital Health’s publication criteria? Is the manuscript technically sound, and do the data support the conclusions? The manuscript must describe methodologically and ethically rigorous research with conclusions that are appropriately drawn based on the data presented.

Reviewer #1: Partly

Reviewer #2: Yes

Reviewer #3: Yes

2. Has the statistical analysis been performed appropriately and rigorously?

Reviewer #1: Yes

Reviewer #2: Yes

Reviewer #3: Yes

3. Have the authors made all data underlying the findings in their manuscript fully available (please refer to the Data Availability Statement at the start of the manuscript PDF file)?

Reviewer #1: No

Reviewer #2: Yes

Reviewer #3: Yes

4. Is the manuscript presented in an intelligible fashion and written in standard English?

Reviewer #1: Yes

Reviewer #2: Yes

Reviewer #3: Yes

5. Review Comments to the Author

Reviewer #1: The authors present a simple and low-cost quantitative gait analysis system for clinical applications.

Although the intent is very commendable, in my opinion the results do not support the clinical usability of the system.

These are the main reasons:

- in comparison with the literature, it is not enough to talk about the costs and simplicity of the protocol and its implementation, but it is necessary to carefully evaluate and compare the reliability, accuracy and repeatability of the system performance, which does not appear in this manuscript.

- methodologically, the control group must perform the same protocol as the experimental group, with the same experimental apparatus and setting. The same goes for data pre-processing.

- in data processing, corrections have been made by visual inspection of the data (not automatic), which is typically time-consuming. An automated processing should be implemented, with a view to having a truly simple and easily usable system in the clinic, as stated by the authors.

- in the presentation of the methods, some offsets were found between the reference instrumentation and the one tested by the authors; the offset has been removed in the processing, but there is no exhaustive justification for the presence of such an offset.

Reviewer #2: Appreciate if the authors can mention what is the level of agreement with standard marker-based motion capture systems, instrumented gait mats .

This is a well written article with simple explanation

Reviewer #3: In this manuscript, the authors demonstrated a valuable study to perform clinical relevant gait analysis with simple videos from two tablets cameras. The background and advances of this study were very clearly reported and so were the experimental design and results. Thus, I recommend acceptance of this work after minor revision based on the following comments.

In Fig.1C, there is a major difference of adding or not adding low-pass filter in the sagittal workflow and frontal workflow, what is the reason of this difference?

The authors mentioned that the unimpaired study data was from a public dataset, which has a different camera setup with the one used in the study. Is there a specific consideration on utilizing the public dataset instead of collecting user data under the same front + sagittal camera setup?

6. PLOS authors have the option to publish the peer review history of their article (what does this mean?). If published, this will include your full peer review and any attached files.

**Do you want your identity to be public for this peer review?** For information about this choice, including consent withdrawal, please see our Privacy Policy.

Reviewer #1: No

Reviewer #2: Yes: Shibu Vijayan

Reviewer #3: No

---

## [Decision Letter · Decision Letter 1]

10 Nov 2023

PDIG-D-23-00173R1

Clinical gait analysis using video-based pose estimation: multiple perspectives, clinical populations, and measuring change

PLOS Digital Health

Dear Dr. Roemmich,

Thank you for submitting your manuscript to PLOS Digital Health. After careful consideration, we feel that it has merit but does not fully meet PLOS Digital Health's publication criteria as it currently stands. Therefore, we invite you to submit a revised version of the manuscript that addresses the points raised during the review process.

Please submit your revised manuscript within 30 days Dec 10 2023 11:59PM. If you will need more time than this to complete your revisions, please reply to this message or contact the journal office at digitalhealth@plos.org. Please include the following items when submitting your revised manuscript:

We look forward to receiving your revised manuscript.

Kind regards,

Mengling Feng

Academic Editor

PLOS Digital Health

Journal Requirements:

2. We ask that a manuscript source file is provided at Revision. Please upload your manuscript file as a .doc, .docx, .rtf or .tex.

3. Please provide separate figure files in .tif or .eps format only and remove any figures embedded in your manuscript file. Please also ensure that all files are under our size limit of 10MB.

Additional Editor Comments (if provided):

Reviewers' comments:

Reviewer's Responses to Questions

**Comments to the Author**

1. If the authors have adequately addressed your comments raised in a previous round of review and you feel that this manuscript is now acceptable for publication, you may indicate that here to bypass the “Comments to the Author” section, enter your conflict of interest statement in the “Confidential to Editor” section, and submit your "Accept" recommendation.

Reviewer #4: (No Response)

2. Does this manuscript meet PLOS Digital Health’s publication criteria? Is the manuscript technically sound, and do the data support the conclusions? The manuscript must describe methodologically and ethically rigorous research with conclusions that are appropriately drawn based on the data presented.

Reviewer #4: Yes

3. Has the statistical analysis been performed appropriately and rigorously?

Reviewer #4: Yes

4. Have the authors made all data underlying the findings in their manuscript fully available (please refer to the Data Availability Statement at the start of the manuscript PDF file)?

Reviewer #4: No

5. Is the manuscript presented in an intelligible fashion and written in standard English?

Reviewer #4: Yes

6. Review Comments to the Author

Reviewer #4: I thank the authors for the opportunity to read this work. I was not involved in the first round of revisions, and as such, do not have any major concerns after reading the revised manuscript and the response to reviewers. Below are a few relatively minor issues that the authors might consider when finalizing the paper for publication.

Introduction, l47: What does the term “data-limited” mean?

l72: I suggest clarifying that the ‘changes in gait’ referred to here specifically relate to changes that occur in response to a change in speed. The reader might otherwise infer that this refers to changes over the course of a longer-term intervention.

l80 (starting “We demonstrate”): This is possibly a style issue, but in my view this part of the introduction actually belongs more in the results or discussion section.

l83 (also l307): It seems odd to refer to tablets as “household video recording devices” or “common household devices”. Why not low-cost cameras or something similar?

Results, l92-103: This is basically a restating of the aims and methods. No actual results are presented in this section. Is it needed?

l119-127 and 128-138: Here again there is a lot of repetition from the methods, with some reference to figures/tables. I wonder if the text could be streamlined somehow to minimize this repetition?

l312-3: I think this sentence is misleading because the kind of changes that occur during recovery or rehabilitation do so over a much longer time course than what was measured here, which was a change in speed within a single session. Rehab-induced changes might also be much more subtle, e.g. the change in preferred speed over a 3-month period. I suggest rewording the sentence to something like: “4) that measure within-participant changes in gait as a result of changes in walking speed”. 

l31-23: This sentence largely repeats the previous paragraph. 

l593-4: This is a good point. Reasons for this are expounded in (at least) the following papers:

- https://www.sciencedirect.com/science/article/pii/S0021929021002402

- https://arxiv.org/pdf/1907.10226.pdf

7. PLOS authors have the option to publish the peer review history of their article (what does this mean?). If published, this will include your full peer review and any attached files.

**Do you want your identity to be public for this peer review?** For information about this choice, including consent withdrawal, please see our Privacy Policy. 

Reviewer #4: No

---

## [Decision Letter · Decision Letter 2]

12 Feb 2024

Clinical gait analysis using video-based pose estimation: multiple perspectives, clinical populations, and measuring change

PDIG-D-23-00173R2

Dear Dr. Roemmich,

We are pleased to inform you that your manuscript 'Clinical gait analysis using video-based pose estimation: multiple perspectives, clinical populations, and measuring change' has been provisionally accepted for publication in PLOS Digital Health.

Best regards,

Mengling Feng

Academic Editor

PLOS Digital Health

Reviewer Comments (if any, and for reference):

Reviewer's Responses to Questions

**Comments to the Author**

1. If the authors have adequately addressed your comments raised in a previous round of review and you feel that this manuscript is now acceptable for publication, you may indicate that here to bypass the “Comments to the Author” section, enter your conflict of interest statement in the “Confidential to Editor” section, and submit your "Accept" recommendation.

Reviewer #4: All comments have been addressed

2. Does this manuscript meet PLOS Digital Health’s publication criteria? Is the manuscript technically sound, and do the data support the conclusions? The manuscript must describe methodologically and ethically rigorous research with conclusions that are appropriately drawn based on the data presented.

Reviewer #4: Yes

3. Has the statistical analysis been performed appropriately and rigorously?

Reviewer #4: Yes

4. Have the authors made all data underlying the findings in their manuscript fully available (please refer to the Data Availability Statement at the start of the manuscript PDF file)?

Reviewer #4: Yes

5. Is the manuscript presented in an intelligible fashion and written in standard English?

Reviewer #4: Yes

6. Review Comments to the Author

Reviewer #4: I thank the authors for their response, and congratulate them on an excellent piece of work

7. PLOS authors have the option to publish the peer review history of their article (what does this mean?). If published, this will include your full peer review and any attached files.

**Do you want your identity to be public for this peer review?** For information about this choice, including consent withdrawal, please see our Privacy Policy.

Reviewer #4: No
